# Human Health Risk Distribution and Safety Threshold of Cadmium in Soil of Coal Chemical Industry Area

**Kai Zhang \*, XiaoNan Li, ZhenYu Song, JiaYu Yan, MengYue Chen and JunCheng Yin**

School of Chemistry and Environment Engineering, China University of Mining and Technology (Beijing), Ding 11, Xueyuan Road, Haidian District, Beijing 100083, China; lxn1110@163.com (X.L.); 20047914@chnenergy.com.cn (Z.S.); 1910380329@student.cumtb.edu.cn (J.Y.); 1910380301@student.cumtb.edu.cn (M.C.); y18811528090@163.com (J.Y.)
\* Correspondence: zhangkai@cumtb.edu.cn; Tel.: +010-62339810

**Abstract:** Cadmium (Cd) is a highly carcinogenic metal that plays an important role in the risk management of soil pollution. In this study, 153 soil samples were collected from a coal chemical plant in northwest China, and the human health risks associated with Cd were assessed through multiple exposure pathways. Meanwhile, by the Kriging interpolation method, the spatial distribution and health risks of Cd were explored. The results showed that the average concentration of Cd in the soil was 0.540 mg/kg, which was 4.821 and 5.567 times that of the soil background value in Ningxia and China, respectively. In comparison, the concentration of Cd in the soil was below the national soil environmental quality three-level standard (1.0 mg/kg). In addition, health risk assessment results showed that the total carcinogenic risk of Cd was $1.269 \times 10^{-6}$–$2.189 \times 10^{-6}$, both above the acceptable criteria ($1 \times 10^{-6}$), while the hazard quotient was within the acceptable level. Oral intake and ingestion of soil particles were the main routes of exposure, and the carcinogenic risk control value of oral intake was the lowest (0.392 mg/kg), which could be selected as the strict reference of the safety threshold for Cd in the coal chemical soil. From Kriging, a prediction map can be centrally predicted on heavy metal pollution in the area surrounding the coal entrance corridor and pedestrian entrance. This study can provide a theoretical basis for the determination of the heavy metal safety threshold of the coal chemical industry in China.

**Keywords:** health risk assessment; coal chemical plant; Cd; soil; safety threshold; Kriging

## 1. Introduction

The development of the modern coal chemical industry has accelerated the clean and efficient utilization of coal resources, especially in coal gasification [1–3]. In recent years, the development of the modern coal chemical industry is mainly concentrated in South Africa, the United States, and China [4]. However, the coal gasification process is often accompanied by the transfer and transformation of heavy metals and other harmful trace elements [5]. Because of the nonbiodegradability, toxicity, and cumulative properties of heavy metals, they can cause environmental pollution and threaten human health after being transferred to the soil, groundwater, air, and other environmental media [6,7]. From 2005 to 2013, the Ministry of Environmental Protection (MEP) and the Ministry of Land and Resources (MLR) of the People's Republic of China jointly carried out the first national soil pollution survey [8], but the overall pollution situation was relatively serious. According to the data of the National Soil Pollution Survey Bulletin in 2014, the standard exceeding ratios of soil heavy metal pollution reached 16.1% in total [9]. Among them, the standard exceeding ratio of cadmium (Cd) was 7.00%, and the proportion of heavy pollution points was 0.50% [10]. In China, soils in more than 11 provinces and 25 regions are rich in Cd [11]. As a worldwide environmental problem, Cd was listed in seventh place as a toxic substance of concern by the American Agency for Toxic Substances and Disease Registry (ATSDR). Moreover, it was listed as a highly toxic, hazardous, and carcinogenic substance

by the European Union [12]. Research shows that long-term exposure to soil environments with high Cd content leads to skeletal damage, renal failure, reproductive effects, and cancers [13,14]. Therefore, it is extremely important to explore soil Cd pollution and assess its health risks to the human body.

The health risk assessment of Cd is crucial, as it can provide valuable information for regional risk management, minimize environmental risk, and protect human health. The current literature on Cd in the soil environment mainly focuses on cities [15–17] and agriculture [18,19]. However, due to the lack of statistical data on heavy metal pollution in emerging industrial sites such as coal chemical industries, there are few studies on soil Cd pollution. In addition, heavy metal fine particles produced by human activities can be redistributed in the environment through wind and atmospheric transport [20]. Through inhalation and skin contact, the human body can be exposed to fine particles containing a variety of heavy metals. Besides occupational exposure, ingestion of contaminated food or water may also be a main way for heavy metals to enter human body [21–23]. At present, the human health risk assessment of heavy metals in soil is mainly based on the model recommended by the US Environmental Protection Agency (EPA), but the toxicological data and exposure parameters of the model are not consistent with the actual characteristics of human exposure in China. Considering the environment and residents' living habits in China, the selection of health exposure parameters suitable for Chinese people can provide theoretical support for the assessment of human health risk. Therefore, taking the coal gasification plant as the research area, based on the health exposure parameters of human body in China, the health risk of Cd in the soil to human body was studied through multiple ways in order to provide a reference for heavy metal ecosystem management in the process of regional planning of the coal chemical industry.

It is a great challenge to accurately simulate the spatial distribution of Cd in soil due to its complex pollution causes [24]. Spatial interpolation is an important method to simulate the spatial distribution of heavy metals in soil. The Kriging interpolation method of the GIS spatial model was introduced into the research of risk assessment of soil heavy metals, and the risk level distribution map of soil heavy metals was obtained, which visually showed the spatial distribution [25]. Because it is the most simple and effective spatial interpolation method, the Kriging interpolation method has been widely used in recent years [26]. The production units in coal chemical plants form a community that is geographically closely connected, and production activities are related to each other. Therefore, the carcinogenic and noncarcinogenic risk distribution of Cd in the soil of the plant area to human health is affected by the distribution of each production unit and the actual production process. Using geostatistics and GIS methods to fit the spatial distribution of Cd carcinogenic and noncarcinogenic risk, a comprehensive evaluation was done for the human health risk assessment of Cd in the plant area, which is helpful to the protection of workers in each production unit.

This paper took a coal chemical plant in northwest China as the research object. Based on the Technical Guidelines for Risk Assessment of Contaminated Sites [27], the human health risk assessment of Cd in the soil in the study area was evaluated. The Kriging interpolation method was used to deeply process the evaluation results. On this foundation, combined with the process production line, the distribution map of human health risk assessment in the plant area was obtained, so that the Cd hazards in different production units were accurately evaluated and their risk sources were analyzed. Most coal chemical plants occupy a large area. A reasonable and accurate evaluation of the pollution degree and health risk can provide an effective and beneficial reference value for land planning and utilization as well as providing a theoretical basis for reducing the harm of Cd on human health in coal chemical sites for further study.

## 2. Materials and Methods

### 2.1. Study Area

The study area is a modern coal chemical plant in northwest China, which is located in Zone A of an energy chemical base. It has a typical continental monsoon climate, with an annual average temperature of 9.0 °C. Southerly wind prevails in summer and northerly wind prevails in winter. The surface water around the plant area is deficient. The annual average precipitation is 199.5 mm, and the annual evaporation is 1752.6 mm, much larger than the former [28]. The physical weathering and wind force are significant. In terms of vegetation, arid grassland is rare in this region. The main soil types are calcareous soil and aeolian sandy soil with poor development. Many days with strong winds exist in the winter and spring dry seasons. All of the above factors are likely to cause the migration of the surface soil polluted by heavy metals and then expand the hazardous area of soil heavy metals pollution.

The plant area occupies 400,000 square meters. It mainly consists of the following areas: a complete modification unit, a variable power distribution unit, an air separation and pressure unit, a water treatment unit, a methanol synthesis unit, a dimethyl ether synthesis unit, a gasification unit, a power unit, a product storage unit, a coal entry corridor, a railway loading/unloading area, two slag dumps, a chimney, a road for staff entry and exit, a road for logistics entry and exit, and main roads in the factory area. The geographical location of the study area and the regional distribution of the plant are shown in Figure 1.

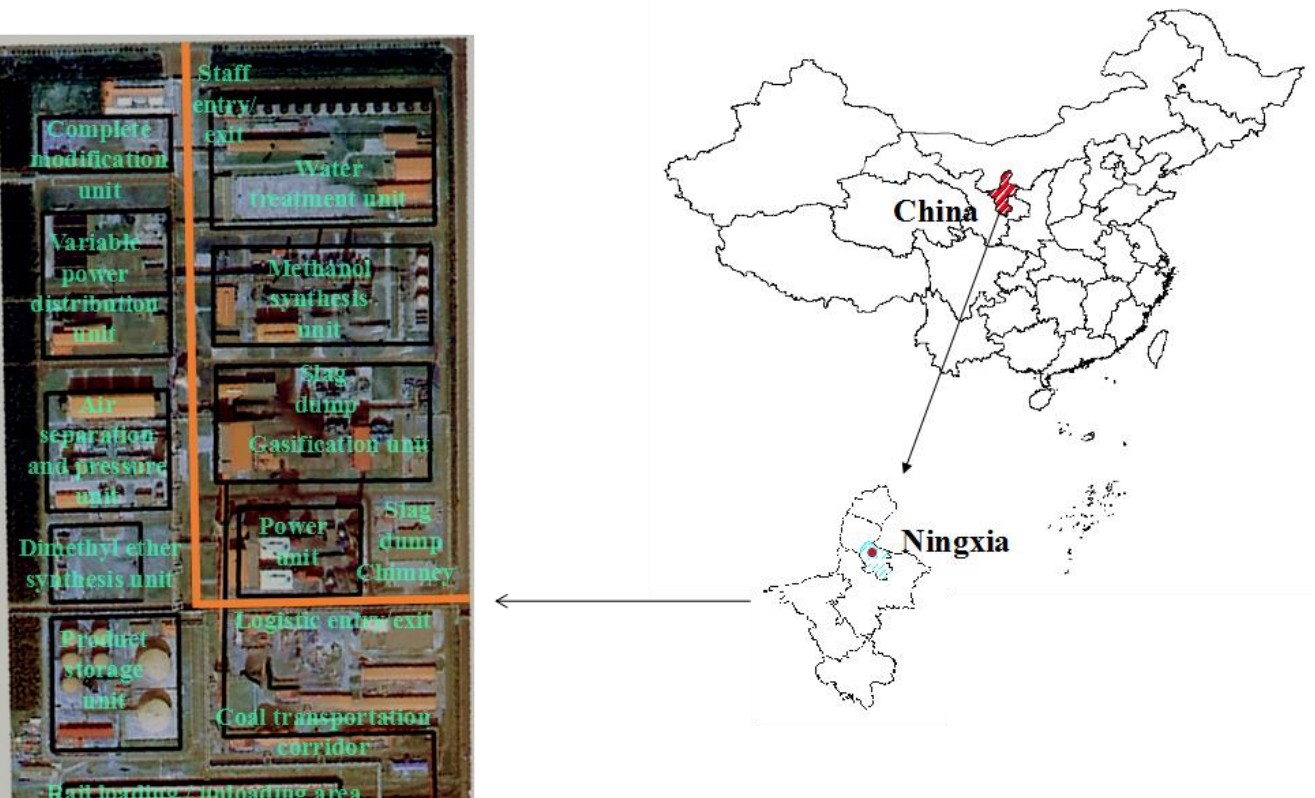

**Figure 1.** Location and regional distribution map of the plant.

### 2.2. Soil Sampling and Analysis

To comprehensively and accurately evaluate the pollution degree of soil heavy metals at different locations in the plant, regional soil samples were collected through a 50 m × 50 m checkerboard distribution method in this study. A total of 153 soil samples from the surface layer (0–20 cm) were collected. In the process of sampling, some sample

points in pools, workshops, slag piles, and other areas, were detrimental to soil collection. Therefore, it was necessary to make appropriate adjustments according to the actual environment around the preset sampling points. Sampling information is shown in Figure 2. The collected soil samples were packed in sealed polyethylene bags, labeled, sent to the laboratory, and dried in a natural environment. After picking out the stone, animal, and plant debris, the samples were ground in an agate mortar and passed through a 100-mesh nylon sieve.

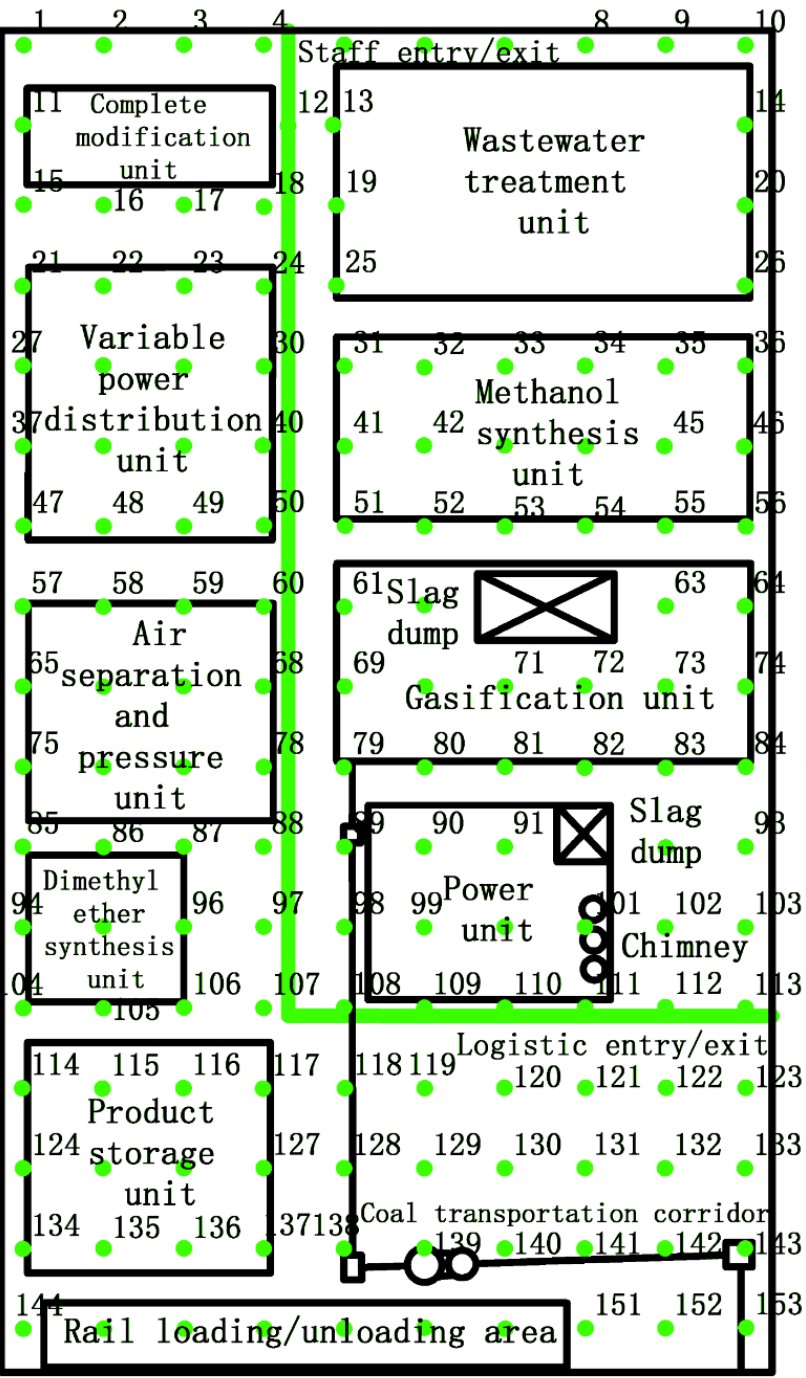

**Figure 2.** Distribution of the sampling points in the coal plant.

The content of Cd in soil was determined by graphite furnace atomic absorption spectrometry with an AAS-ZEEnit 700 Atomic Absorption Spectrometer (Jena, Germany).

The detection limit was 0.006 mg/kg. A 0.300 g sample was accurately weighed and put into a beaker. Ten milliliters of concentrated $HNO_3$, 4ml 3% (mass fraction) $H_2O_2$, and 10 mL HF were added in turn for microwave digestion. After digestion, 1% (mass fraction) $HNO_3$ was added to a 50 mL colorimetric tube for standby (GB/T 17141–1997).

*2.3. Quality Control*

To control quality, each batch of samples used the same reagents and followed the same steps to make 2 reagent blanks. Into each batch of samples was added a nationally certified reference material (GSS–10 and GSS–11 from the Ministry of Environmental Protection Standard Sample Institute). The relative standard deviation was usually within $\pm 2\%$; otherwise, the sample was measured again. The measurement results of the blank samples were all less than the detection limit of the method, and the recovery rate of the matrix spiked was in the range of 80–130% [29].

*2.4. Human Health Risk Assessment*

The Ministry of Environmental Protection of the PRC has released the Technical Guidelines for Risk Assessment of Soil Contamination of Land for Construction [27]. The human health risk assessment models recommended in the Technical Guidelines are based on the EPA models. However, in order to reflect the actual contamination situations in China, a suitable parameter is given according to the environment and living habits of Chinese residents. Based on the pollution characteristics of coal chemical sites, the site type was verified, the exposure route selected, and the exposure amount, carcinogenic risk, hazard quotient, and risk control value calculated. On this basis, the contribution rate of Cd carcinogenic risk under different exposure routes was analyzed, and the environmental safety threshold of Cd in the soil of the study area was determined.

2.4.1. Exposure Pathways

The study site is a coal chemical industry area, a non-insensitive industrial land, with no surface water around and no groundwater as drinking water. In this paper, oral intake, skin contact, and inhalation of soil particles were selected as the main exposure routes to conduct a human health risk assessment of Cd. The soil exposure model (Table 1) and exposure factor parameters (Table 2) corresponding to carcinogenic and noncarcinogenic effects of a single pollutant were selected [25]. There are no children in the coal chemical plant area, so children's exposure was not considered, and only adults were selected as recipients.

**Table 1.** Calculating models of soil exposure dose in three soil exposure routes.

| Exposure Routes | Explanation | Formula Expression of Exposure |
|---|---|---|
| Mouth intake | Carcinogenic | $\text{OISER}_{ca} = \frac{\text{OSIR}_a \times \text{ED}_a \times \text{EF}_a \times \text{ABS}_0}{\text{BW}_a \times \text{AT}_{ca}} \times 10^{-6}$ |
| | Noncarcinogenic | $\text{OISER}_{nc} = \frac{\text{OSIR}_a \times \text{ED}_a \times \text{EF}_a \times \text{ABS}_0}{\text{BW}_a \times \text{AT}_{nc}} \times 10^{-6}$ |
| Skin contact | Carcinogenic | $\text{DCSER}_{ca} = \frac{\text{SAE}_a \times \text{SSAR}_a \times \text{EF}_a \times \text{ED}_a \times \text{E}_V \times \text{ABS}_d}{\text{BW}_a \times \text{AT}_{ca}} \times 10^{-6}$ |
| | Noncarcinogenic | $\text{DCSER}_{nc} = \frac{\text{SAE}_a \times \text{SSAR}_a \times \text{EF}_a \times \text{ED}_a \times \text{E}_V \times \text{ABS}_d}{\text{BW}_a \times \text{AT}_{nc}} \times 10^{-6}$ |
| Inhalation of soil particles | Carcinogenic | $\text{PISER}_{ca} = \frac{\text{PM}_{10} \times \text{DAIR}_a \times \text{ED}_a \times \text{PIAF} \times (\text{fspo} \times \text{EFO}_a + \text{fspi} \times \text{EFI}_a)}{\text{BW}_a \times \text{AT}_{ca}} \times 10^{-6}$ |
| | Noncarcinogenic | $\text{PISER}_{nc} = \frac{\text{PM}_{10} \times \text{DAIR}_a \times \text{ED}_a \times \text{PIAF} \times (\text{fspo} \times \text{EFO}_a + \text{fspi} \times \text{EFI}_a)}{\text{BW}_a \times \text{AT}_{nc}} \times 10^{-6}$ |

Note: In Table 1, $\text{OISER}_{ca}$ soil exposure dose in oral intake (carcinogenic) in milligrams per kilogram per day, $\text{OISER}_{nc}$ soil exposure dose in oral intake (noncarcinogenic) in milligrams per kilogram per day, $\text{DCSER}_{ca}$ soil exposure dose in skin contact (carcinogenic) in milligrams per kilogram per day, $\text{DCSER}_{nc}$ soil exposure dose in skin contact (noncarcinogenic) in milligrams per kilogram per day, $\text{PISER}_{ca}$ soil exposure dose in inhalation (carcinogenic) in milligrams per kilogram per day, $\text{PISER}_{nc}$ soil exposure dose in inhalation (noncarcinogenic) in milligrams per kilogram per day.

**Table 2.** Major parameters in the models for calculating exposure doses.

| Parameter | Definition | Value | Units |
|---|---|---|---|
| $OSIR_a$ | Daily soil intake of adults | 100 | mg day$^{-1}$ |
| $ED_a$ | Adult exposure period | 25 | a |
| $EF_a$ | Adult exposure frequency | 250 | day a$^{-1}$ |
| $BW_a$ | Adult weight | 56.8 | kg |
| $ABS_0$ | Absorption efficiency factor of mouth-intake soil | 1 | - |
| $AT_{ca}$ | Average time of carcinogenesis | 26,280 | day |
| $AT_{nc}$ | Average time of noncarcinogenesis | 9125 | day |
| $SAE_a$ | Surface area of adults' exposed skin | 2854.63 | cm$^2$ |
| $SSAR_a$ | Soil sticking coefficient of adults' skin surface | 0.2 | mg cm$^2$ |
| $ABS_d$ | Skin-contact soil absorption efficiency factor | 0.001 | - |
| $E_v$ | Skin daily contact event frequency | 1 | times day$^{-1}$ |
| $PM_{10}$ | Content of inhalable suspended particulate matter in air | 0.15 | m$^3$ day$^{-1}$ |
| $DAIR_a$ | Adults' daily intake of air | 14.5 | m$^3$ day$^{-1}$ |
| PIAF | The retention ratio of soil particles in body after inhalation | 0.75 | - |
| fspi | The proportion of soil particles in indoor air | 0.8 | - |
| fspo | The proportion of soil particles in outdoor air | 0.5 | - |
| $EFI_a$ | Indoor exposure frequency of adults | 187.5 | day a$^{-1}$ |
| $EFO_a$ | Outdoor exposure frequency of adults | 62.5 | day a$^{-1}$ |
| $C_{sur}$ | Pollutants' concentration in the surface soil | Table 1 | mg kg$^{-1}$ |
| $SF_0$ | Carcinogenic slope factor of mouth-intake soil | 6.1 | mg$^{-1}$ kg day |
| $SF_d$ | Carcinogenic slope factor of skin-contact soil | 1.5 | mg$^{-1}$ kg day |
| $SF_i$ | Carcinogenic slope factor of inhalation | 7.051 | mg$^{-1}$ kg day |
| SAF | Reference dose distribution coefficient exposed to soil | 0.2 | - |
| $RfD_0$ | Reference dose of mouth-intake soil | $1.00 \times 10^{-3}$ | mg kg$^{-1}$ day$^{-1}$ |
| $RfD_d$ | Reference dose of skin-contact soil | $2.50 \times 10^{-5}$ | mg kg$^{-1}$ day$^{-1}$ |
| $RfD_i$ | Reference dose of inhalation | $2.553 \times 10^{-6}$ | mg kg$^{-1}$ day$^{-1}$ |

Note: In Table 2, the exposure parameters only targeted adults.

### 2.4.2. Risk Characterization and Contribution Rate

Human health risk brought about by Cd pollution was characterized by a single pollutant risk. The risk characterization and contribution rate can be calculated by the formula of carcinogenic risk and hazard quotient of different soil exposure routes (Table 3) [30].

**Table 3.** Carcinogenic risk and hazard quotient calculating formulas for three soil exposure routes.

| Exposure Route | Formula Description | Formula Expression |
|---|---|---|
| Oral intake | Carcinogenic risk | $CR_{ois} = OISER_{ca} \times C_{sur} \times SF_o$ |
| | Hazard quotient | $HQ_{ois} = \dfrac{OISER_{nc} \times C_{sur}}{RfD_O \times SAF}$ |
| Skin contact | Carcinogenic risk | $CR_{dcs} = DCSER_{ca} \times C_{sur} \times SF_d$ |
| | Hazard quotient | $HQ_{dcs} = \dfrac{DCSER_{nc} \times C_{sur}}{RfD_d \times SAF}$ |
| Inhalation of soil particles | Carcinogenic risk | $CR_{pis} = PISER_{ca} \times C_{sur} \times SF_i$ |
| | Hazard quotient | $HQ_{pis} = \dfrac{PISER_{nc} \times C_{sur}}{RfD_i \times SAF}$ |

Note: In Table 3, $CR_{ois}$—carcinogenic risk of mouth-intake soil; $CR_{dcs}$—carcinogenic risk of skin-contact soil; $CR_{pis}$—carcinogenic risk of soil particle inhalation; $HQ_{ois}$—hazard quotient of mouth-intake soil; $HQ_{dcs}$—hazard quotient of skin-contact soil; $HQ_{pis}$—hazard quotient of soil particle inhalation.

Based on the Cd pollution risk assessment, the contribution rates to the risk of the three different routes were calculated (Formula (1)), and the main route was analyzed to obtain evidence for prevention and control of the subsequent risk.

$$R_i = \frac{CR_i}{\sum CR_i} \times 100\% \tag{1}$$

In the above formula: $CR_i$ means the carcinogenic risk contribution rate or hazard quotient level of a certain exposure route, and the dimension is 1; $\Sigma CR_i$ means the total carcinogenic risk or total hazard quotient.

### 2.4.3. Risk Control Thresholds

The acceptable risk value of carcinogens defined by the USEPA is that the lifetime risk of cancer exceeds the normal value of $1 \times 10^{-4}$–$1 \times 10^{-6}$. When pollutant hazard exceeds the acceptable level of human carcinogenic risk ($1 \times 10^{-6}$) or hazard quotient (1), the risk control value should be calculated corresponding to its soil exposure route. The calculation formula of the safety threshold of three kinds of soil exposure paths can be shown in Table 4 [30].

**Table 4.** Safety threshold calculating formulas for three soil exposure routes.

| Exposure Route | Formula Description | Safety Threshold Calculating Formulas |
|---|---|---|
| Mouth-intake soil | Carcinogenic risk | $RCVS_{ois} = \frac{ACR}{OISER_{ca} \times SF_0}$ |
| | Hazard quotient | $HCVS_{ois} = \frac{RfD_0 \times SAF \times AHQ}{OISER_{nc}}$ |
| Skin-contact soil | Carcinogenic risk | $RCVS_{dcs} = \frac{ACR}{DCSER_{ca} \times SF_d}$ |
| | Hazard quotient | $HCVS_{dcs} = \frac{RfD_d \times SAF \times AHQ}{DCSER_{nc}}$ |
| Inhalation of soil particles | Carcinogenic risk | $RCVS_{pis} = \frac{ACR}{PISER_{ca} \times SF_i}$ |
| | Hazard quotient | $HCVS_{pis} = \frac{RfD_i \times SAF \times AHQ}{PISER_{nc}}$ |

Note: In Table 4, RCVSois—soil safety threshold based on carcinogenic effect of mouth-intake soil pathways, mg/kg; ACR—acceptable carcinogenic risk, which is dimensionless and takes the value of $10^{-6}$; AHQ—acceptable hazard quotient, which is dimensionless and takes the value of 1; RCVSdcs—soil safety threshold based on carcinogenic effect of skin-contact soil pathways, mg/kg; RCVSpis—soil safety threshold based on carcinogenic effect of soil particle inhalation pathways, mg/kg.

### 2.5. Kriging Interpolation Method

The Kriging interpolation of the GIS spatial model is a geostatistical method that is used in smoothing surfaces and predicting the values of unsampled locations. It provides the best linear unbiased estimation and estimation error distribution information, which shows a strong statistical advantage [31,32]. Its formula is expressed as [33]:

$$Z_{(x_0)} = \sum_{i=1}^{n} \lambda_i Z_{(x_i)} \tag{2}$$

In the Formula (2), $Z_{(x0)}$ represents the concentration of heavy metal in the estimation points, and $Z_{(xi)}$ represents the concentration of heavy metal in the sampling point *i*. *n* represents the total number of sampling points, and $\lambda_i$ is a set of weight coefficients, the value of which is to ensure minimum variance of the result and an unbiased estimate.

## 3. Results and Discussion

### 3.1. The Descriptive Statistics of Cd

The statistical characteristic values of Cd content in the soil of the coal chemical plant area are listed in Table 5. It was measured that the concentration of Cd in the soil of the coal chemical factory was 0.400~0.690 mg/kg, with an average concentration of 0.540 mg/kg. According to the total Cd concentration standard of 20 mg/kg (industrial land) in the soil environmental quality standard GB15618-2008 [34], the Cd concentration did not exceed the standard. However, compared to the national background value of 0.097 mg/kg and the Ningxia background value of 0.112 mg/kg, the rate of exceeding the standard in all production units reached 100%. From the spatial distribution of Cd content in the soil, the coefficient of variation was small, with an average value of 11.043%, indicating that the spatial distribution of Cd in the soil was uniform and the degree of dispersion was small.

**Table 5.** Measurement and statistics of soil sample Cd content.

| Concentration (mg/kg) | Ningxia Background Value [a] (mg/kg) | Percentage Exceeded Based on Ningxia | National Background Value [a] (mg/kg) | Percentage Exceeded Based on China | Average (mg/kg) | Standard Deviation | Variation Coefficient % |
|---|---|---|---|---|---|---|---|
| 0.400~0.690 | 0.112 | 100% | 0.097 | 100% | 0.540 | 0.060 | 11.043 |

[a] Data from China Environmental Monitoring Station, 1990.

### 3.2. Variogram Model Fitting

3.2.1. Spatial Autocorrelation Test

The spatial autocorrelation (Moran's I) tool of ArcGIS spatial statistics was used to analyze the carcinogenic risk and hazard quotient of Cd pollution. The results are shown in Table 6. It can be seen from Table 6 that the value of Moran's I was relatively close to 1, so the hazard quotient of Cd was a significant cluster with a positive correlation [35].

**Table 6.** Digital features of autocorrelation in data space.

| Element | Moran's I | *z* Value | *p* Value |
|---|---|---|---|
| Cd | 0.814 | 12.072 | 0.001 |

By the criterion of spatial autocorrelation, compared with the *z* value and *p* value in Table 6, it was indicated that the spatial autocorrelation was very significant. Therefore, simulation analysis could be used in the spatial interpolation method of Kriging.

3.2.2. Analysis of Spatial Structure Variation

Semivariance analysis is the key to spatial variation structure analysis and plays an important role in reasonably controlling the number of sampling points and improving the accuracy of interpolation. When choosing the semivariance function model, the closer the error standard average of the cross-validation result is to 0, the closer the root mean square of error standard is to 1, which means the higher precision of the simulation of the model [36]. Table 7 showed the fitting results of the semivariance model of total carcinogenic risk and total hazard quotient of Cd in Kriging interpolation points.

**Table 7.** Fitting results of the semivariogram model.

| Evaluation Approach of Cd | Model | Nugget Constant $C_0$ | Sill Value | ($C_0$/Sill)/% | Variable Course/m | Standard Error of the Mean | Root Mean Squared Error | K–S Test *p* Value | Coefficient of Determination/$R^2$ |
|---|---|---|---|---|---|---|---|---|---|
| Total carcinogenesis | Gauss | 0.002 | 0.006 | 33.330% | 103.856 | −0.004 | 1.007 | 0.137 | 0.967 |
| Total noncarcinogenesis | Gauss | 2.322 | 6.703 | 34.640% | 103.856 | −0.004 | 1.007 | 0.158 | 0.885 |

From the ratio of nugget value $C_0$ to the sill value ($C_0$/Sill), the interpolated value of Cd was ranged from 33.3 to 34.64% and between 25 and 75% (Table 7), which was a medium correlation, indicating that the influence of regional factors on the variation of soil Cd content in the chemical plant was significantly greater than that of nonregional factors (i.e., random factors).

### 3.3. Human Health Risk Assessment of Soil Cd in Coal Chemical Area

The risk values of soil Cd in the coal chemical industry area were calculated according to oral intake, skin contact, and inhalation of soil particles. The total carcinogenic risk value and total risk quotient were calculated by summing. The results are shown in Table 8.

**Table 8.** The results of risk of Cd in the coalification area.

| Cd | Types | Oral Intake Soil | | Skin Contact Soil | | Inhalation of Soil Particle | | Total Carcinogenic Risk | Total Hazard Quotient |
| | | Carcinogenic Risk | Hazard Quotient | Carcinogenic Risk | Hazard Quotient | Carcinogenic Risk | Hazard Quotient | | |
|---|---|---|---|---|---|---|---|---|---|
| Overall N = 153 | Min | $1.022 \times 10^{-6}$ | 0.0024 | $2.333 \times 10^{-7}$ | 0.0006 | $1.397 \times 10^{-8}$ | 0.0156 | $1.269 \times 10^{-6}$ | 0.014 |
| | Max | $1.762 \times 10^{-6}$ | 0.0042 | $4.025 \times 10^{-7}$ | 0.0010 | $2.409 \times 10^{-8}$ | 0.0193 | $2.189 \times 10^{-6}$ | 0.024 |
| | Average | $1.410 \times 10^{-6}$ | 0.003 | $3.230 \times 10^{-7}$ | 0.0010 | $1.930 \times 10^{-8}$ | 0.0150 | $1.754 \times 10^{-6}$ | 0.020 |

### 3.3.1. Carcinogenic Risk

The Kriging interpolation method was used to simulate the multipath total carcinogenic risk value of Cd in the soil in Table 8, and the total carcinogenic risk assessment map of the whole plant area was obtained (Figure 3). According to the results in Table 8 and Figure 3, the multipath combined carcinogenic risk value of Cd in the soil area of the chemical unit in the plant area was in the range of $1.269 \times 10^{-6}$–$2.189 \times 10^{-6}$, with an average value of $1.754 \times 10^{-6}$, which exceeded the acceptable level of human health ($1 \times 10^{-6}$).

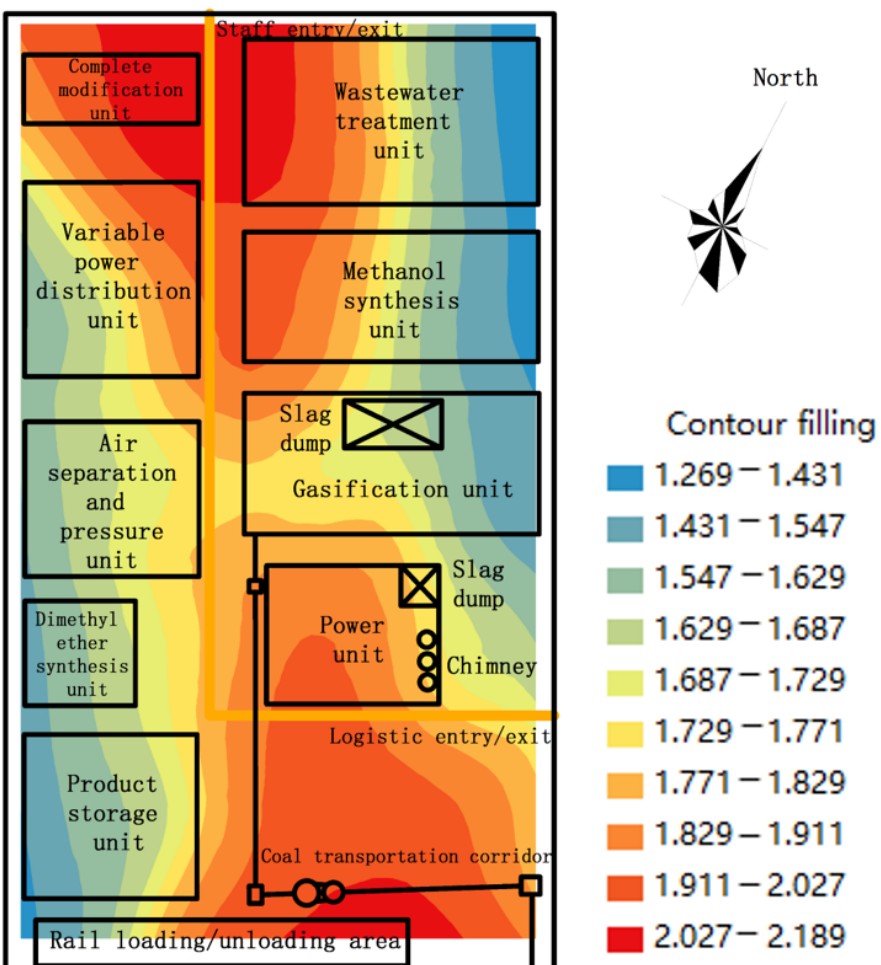

**Figure 3.** Cd total carcinogenic risk.

It can also be seen from the figure that the spatial distribution of the pollution area was uneven. The high pollution area was mainly divided into two parts. One part was near the coal entry corridor. Mainly heavy-duty transport vehicles pass through the road system in the mining area, producing a large amount of dust, and the transported mineral materials may be dispersed to a certain extent, constituting an important source of heavy metals, especially lead, nickel, and Cd [37]. The content of heavy metals in soils located along roads is strongly related to traffic and decreases with increasing distance from the road [38].

Cd pollution in the soil around the coal mine was caused by coal combustion, coal mining, the coal transportation process, and the migration and settlement of coal gangue and coal dust. This is consistent with our results [39,40].

Another part of the high pollution area was near the entrance and exit for people. The accumulation of Cd in the soil was related to the common effects of coal mineralization and accumulation and human disturbance. Anthropogenic sources account for more than 90% of the Cd released from the environment [41]. Research by Xu et al. showed that Cd pollution levels were relatively high in areas with intensive human activities in the coal mine area while exploring Cd's harm to human health [42]. The risk value decreases from the northwest and southeast to the center and surroundings. There is a chimney in the power unit, so Cd settles down with the smoke and dust, which leads to a higher cancer risk of Cd around the chimney and in the southwest in a phenomenon consistent with plume dispersion patterns as reflected in the regulatory understanding of atmospheric emissions [43]. Regulatory guidance suggests carrying out harmless treatment of the slag of the total transformation unit and the methanol synthesis unit and controlling the emission of flue gas to reduce or block the potential carcinogenic risk.

### 3.3.2. Hazard Quotient

The Kriging interpolation method was used to simulate the multipath total hazard quotient of soil Cd in Table 8, and the hazard quotient evaluation map of the entire plant area was obtained (Figure 4). It can be seen from Table 8 and Figure 4 that the Cd multipath hazard quotients in the soil area of the chemical units were 0.014–0.024 (with an average of 0.020), all of which were within the acceptable level of 1. The main polluted areas were those near the coal transportation corridor, power unit, complete modification unit, and the entry/exit for people.

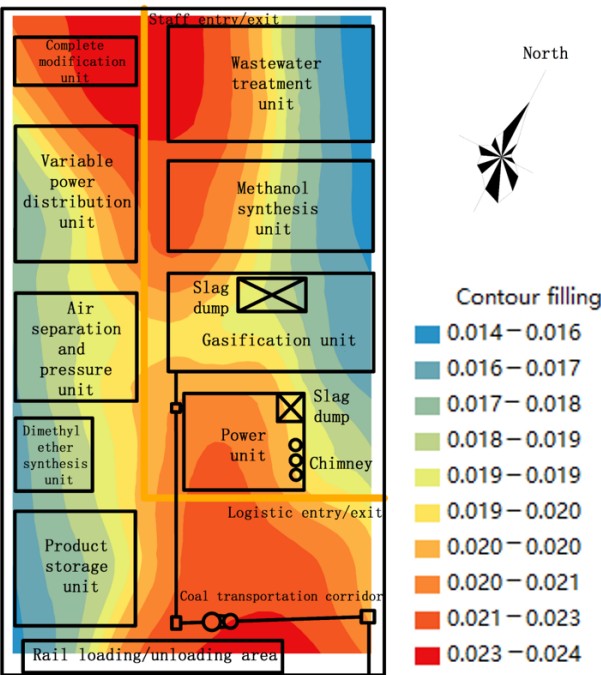

**Figure 4.** Total hazard quotient risk map for Cd.

On the whole, the multipath total hazard quotient of Cd in the soil of the main process unit of coal gasification was relatively low. Therefore, its fluctuation under an acceptable level can be ignored. However, Cd is a common environmental contaminant with a long biological half-life in the soil. Moreover, its microbial or chemical loss is small, so it can exist for a long time in the soil. With the long-term development of the factory, Cd accumulates continuously in the soil [44–46]. Gorospe [47] analyzed 16 different heavy

metals in 91 soil samples from a vegetable garden in San Francisco. The results showed that soils from most (>75%) gardens contained higher levels of heavy metals than those of the California human health screening, and the level of Cd was the highest, reaching 84%. Cd has high plant–soil mobility and enters the human body through the food chain. When it enters the human body, the half-life of Cd can last for 10–30 years, which causes chronic toxicity [48]. Therefore, some attention should be paid to the situation of the entry and exit of people to avoid harm to human health caused by Cd accumulation pollution.

### 3.3.3. Contribution Ratios of Human Health Risk for Different Exposure Routes

The risk contribution rates of carcinogenic risk and hazard quotient under different exposure routes (Figure 5) were calculated to provide the basis for targeted prevention and control of human health risk in coal chemical industry areas. It can be seen from Figure 5 that the carcinogenic risk of Cd in the soil of the coal chemical industry areas was mainly caused by oral intake of soil dust, accounting for 80.51% of the total carcinogenic risk, which was 4 and 73 times of the contribution rates of skin exposure and inhalation of soil particles, respectively. However, for the noncarcinogenic risk of Cd in the coal chemical industry area, the contribution rates of the three exposure pathways of Cd in the soil of the coal chemical industry area were respiratory inhalation (79%) > oral intake (17%) > skin contact (4%). Inhalation of soil particles was the main noncarcinogenic exposure route of Cd. Heavy metals in street dust mainly come from vehicle emissions and coal combustion. Some studies [49,50] have showed that coal transportation has an impact on the environment. Dust is rich in heavy metal pollutants and could be easily inhaled or ingested. Also, the regional evaporation is much greater than the precipitation, so the vegetation is sparse and scattered, and the physical weathering and wind effects are obvious. These factors are likely to lead to a higher carcinogenic and noncarcinogenic risk from inhalation and oral intake.

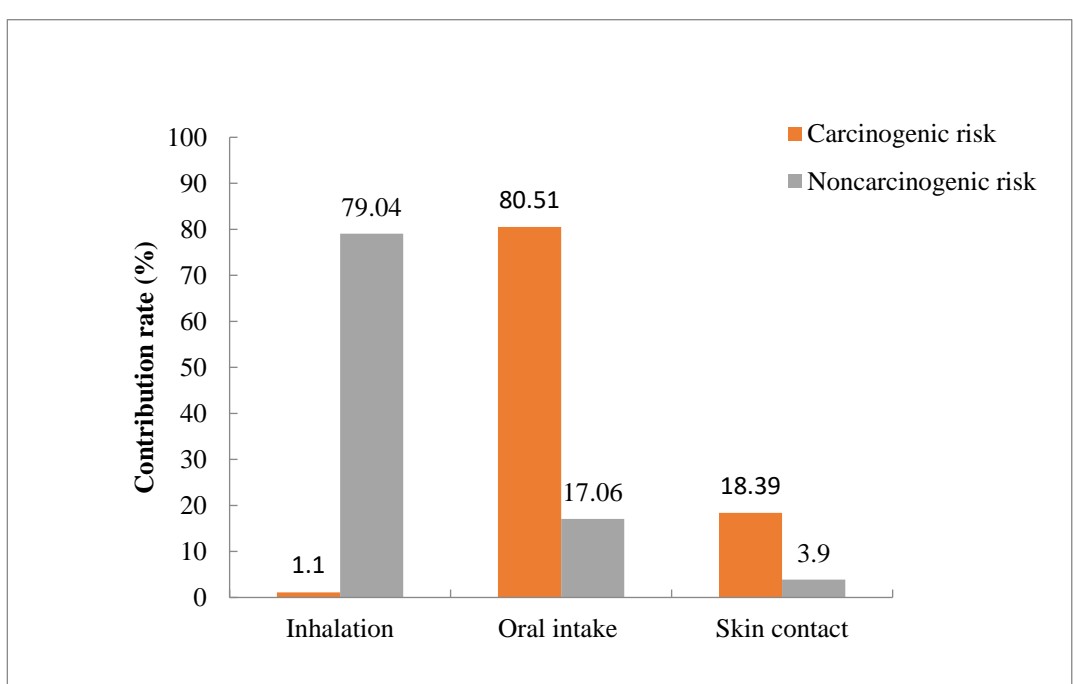

**Figure 5.** Carcinogenic risk and noncarcinogenic risk contribution rate of cadmium from three exposure pathway.

In conclusion, considering the human health risks of Cd in the coal chemical industry area, oral intake and inhalation of soil particles were the main exposure routes, so targeted control and prevention should be carried out during the production process of the coal chemical area. Yang et al. [51] evaluated the health risks of heavy metals in the soil of Changchun and found that Cd was mainly exposed through inhalation and oral intake.

This is consistent with our findings. Therefore, in the production process of the coal chemical industry, it is necessary to prevent the exposure of soil particles by oral intake and inhalation. For workers in the plant area, protective measures should be strengthened to reduce exposure of Cd in the soil due to their high exposure. For example, water can be sprayed regularly in the exposed area of soil in the factory districts to suppress dust, and protective masks can be used, to reduce potential risks.

### 3.4. Safety Threshold of Cd in the Coalification Zone Soil

Based on the human health risk assessment of Cd in the coal chemical industry, the risk control values for different exposure routes were calculated (Table 9). When the hazard quotient does not exceed the standard, the control value can be ignored. As is shown in Table 9, the risk control values of Cd under different exposure routes were different within the acceptable carcinogenic risk level of 1E-06. The risk control value of Cd for oral intake was the minimum (0.392 mg/kg), slightly higher than the soil background value, and far less than the industrial land standard. Therefore, the human health risk assessment based on different exposure pathways can be used as a strict standard for the safety threshold of Cd in the soil environment. Different choices of risk levels will also lead to different safety thresholds. For example, the USEPA recommends $1 \times 10^{-6}$–$1 \times 10^{-4}$, the United Kingdom generally adopts $1 \times 10^{-5}$ in practice, and the Netherlands suggests a looser $1 \times 10^{-4}$ [52]. In addition, the background value, geological conditions, and other data should be comprehensively considered to determine the safety threshold of Cd in the soil environment of certain sections of coal chemical areas, which can provide a comprehensive safety guarantee and guidance for human health and regional development of Cd in China's coal chemical industry field [53].

**Table 9.** Risk control value of Cd in the coalification zone soil.

| Exposure Routes | Risk Types | Risk Control Value (mg/kg) |
|---|---|---|
| Oral-intake soil | carcinogenic | 0.392 |
| Skin-contact soil | carcinogenic | 1.714 |
| Inhalation of soil particles | carcinogenic | 28.641 |

### 4. Conclusions

In this study, the distribution of human health risks and the safety threshold of Cd in the soil of the coal chemical plant area were researched. The Cd content in the soil of coal chemical plants was higher than the regional and national background values. Human health risk assessment revealed that the total carcinogenic risk values of Cd exceeded the acceptable level ($1 \times 10^{-6}$). It is necessary to regularly spray water on the soil exposure areas of the plant to suppress the dust in the plant and to wear protective masks to reduce the potential risk.

The spatial distribution of Cd pollution in the plant area was uneven, mainly distributed in the outflow entrance and coal entrance corridor, which is highly dangerous for health. The hazard quotient of Cd was within the acceptable level, but Cd is liable to accumulate in the soil, so it is necessary to pay close attention to avoid its harmful effects on the human body.

The safety threshold of the plant was taken as 0.392 mg/kg, higher than the soil background value. However, compared with the industrial land exceeding standard, there is still a long way. This standard may be too strict for the determination of the safety threshold of the coal chemical industry area, accounting for the difference in carcinogenic risk level and the selection of parameters in different counties and districts as well as the gap between the high toxicity of Cd to human health. The concentration threshold that poses a risk or hazard to human health is low. Therefore, a series of factors such as regional soil background values, geological environments, and different carcinogenic risk levels should be comprehensively considered to determine the safety threshold of heavy metals in the coal chemical industry area.



**Author Contributions:** K.Z.: outline of the manuscript, regional field investigation, soil sample collection and analysis, data analysis, graph and table drawing, writing the manuscript; X.L.: sample testing, data collation, graph and table drawing, manuscript writing and revision, reference collation, review comments reply; Z.S.: soil sample collection, preparation of some figures and tables, writing the manuscript; J.Y. (JiaYu Yan): data analysis, writing a part of the manuscript (Reference); M.C.: writing a part of the manuscript (Reference). J.Y. (JunCheng Yin): writing a part of the manuscript (Introduction). All authors have read and agreed to the published version of the manuscript.

**Funding:** This work was co-supported by the Research on Ecological Restoration and Protection of Coal Base in Arid Eco-fragile Region (GJNY2030XDXM-19-03.2), National Key Research and Development Program of China (2018YFC0406404), Yue Qi Young Scholar Project, China University of Mining and Technology (Beijing) (2019QN08), the Fundamental Research Funds for the Central Universities (2020YJSHH12), the science and technology project of Shendong company in 2020 (202016000041).

**Acknowledgments:** We wish to thank Zheng, X.H. and Wang, Y.J. for their help in the process of writing the manuscript, thank all reviewers who put forward their opinions on the revision of the manuscript, as well as all editors in charge of this manuscript.

**Conflicts of Interest:** The authors declare no conflict of interest.

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
