# Peer review of "Human Health Risk Distribution and Safety Threshold of Cadmium in Soil of Coal Chemical Industry Area"

_minerals, doi:10.3390/min11070678_

Round 1

Reviewer 1 Report

Dear Authors

In my opinion, the theme of the article is not innovative, but very interesting for the readers of the journal or in a journal of Risk Assessment.

One hundred fifty-three soil samples were collected from a coal chemical plant in northwest China. The spatial distribution and health risk of Cd were studied.

The authors must clearly refer to the soil fraction used.

The distribution of human health risks and the safety threshold of Cd in the soil of the coal chemical plant area were researched.

The Cd content in the soil of coal chemical plants was higher than the regional and national background values, however the average concentration of Cd in the soil is below the national soil environmental quality three-level standard.

The health risk assessment results showed that the total carcinogenic risk of Cd was close to the acceptable criteria (1E-06 to 10-4), while the hazard quotient was within the acceptable level.

Cd is liable to be accumulated in the soil, so it is necessary to pay close attention to avoid its harmful effects on the human body.

As usual, oral intake and ingestion of soil particles were the main routes of exposure.

The authors found that the carcinogenic risk control value of oral intake was the lowest, which could be selected as the strict reference of the safety threshold for Cd in the coal chemical soil.

Kriging showed that the spatial distribution of Cd pollution in the plant area was uneven, mainly distributed in the outflow entrance and coal entrance corridor, which is highly dangerous for health.

This research can provide a theoretical basis for the determination of the potentially toxic elements safety threshold of the coal chemical industry in China.

The manuscript under revision is well structured, the language is correct and clear and the title and abstract clearly describe the content of the manuscript.

In my opinion, the manuscript is almost ready to be published. Please find attached the revised suggestions (minor revisions).

Congratulations!

Best regards

Author Response

Dear teacher,

Thank you for your feedback and allow us to improve our manuscript "Human health risk distribution and safety threshold of cadmium in soil of coal chemical industry area",(minerals-1268382).

We had revised the manuscript according to your comments. References were added to the revised manuscript. In addition, About your question about 100-mesh nylon sieve, 100 mesh is 0.150 mm. Regarding the question of the table in the manuscript, we had revised it in the newly revised manuscript.

Yours sincerely,

Kai Zhang

Name: Kai Zhang

E-mail: zhangkai@cumtb.edu.cn

Reviewer 2 Report

Evaluation

Authors present the analysis of  human health risk caused by Cd in the area of selected coal chemical plant in Northwest China. Both carcinogenic and non-carcinogenic risk was assessed taking into account three pathways of exposure (oral, dermal and inhalation). Kriging interpolation method was used for the spatial distribution of risk associated with Cd concentration in the study area. Finally, the risk control values of Cd were established for three exposure pathways. The manuscript is properly organized and well written. However, the subject of this study seems to be of low novelty and may be interesting mainly for local audience. It is worth outlining the research problem in a broader (international) context.

Specific comments

Lines 93-94: the following sentence is unclear: “By the Kriging interpolation method and process production line, the evaluation results were further processed..” – please rephrase

Line 100: please insert “on” between “Cd” and “human health”

Line 107: please be more precise in the following statement “The surface water around the plant area is poor” – what does “poor water” mean?

Figure 1. the quality of this figure have to be improved, mainly in the part showing the regional distribution map of the plant - it is currently difficult to read.

Lines 129-130: this sentence seems to be unfinished “Offset collection or skipping of locations that are difficult to collect, such as pools, workshops, and slag dumps” – please rephrase

Lines 151-189: Subchapter “Human health risk assessment”: in this chapter both the models and parameters used for carcinogenic and non-carcinogenic risk calculation should be given (in methodological part or in supplementary materials)

Lines 179-181: please carefully check (and correct) explanations for Ri and CRi - the explanations seem to be identical

Lines 191-199: please indicate which statistical tool was used for the kriging analysis and statistical evaluation of Cd content in soils

Line 219: it should be “It can be seen from Table 2 that the value of Morans’ I was…”

Lines 221-222: please change “Table 1” to “Table 2”

Line 232: the results of semi-variance model are given in Table 3 (not in Table 6)

Line 234: It should be “the ratio of nugget value C0 to the sill value (C0/Sill)”

Line 244: please change “consequences” to “results”

Table 4. Please specify what is given in this Table – minimum/maximum or mean/median values

Lines 282-283: “(with an average of 0.020)” – mean values are not given in Table 4 and Figure 4 – please correct results

Figure 3 and 4: text in both Figures is blurry and unreadable – please improve

Line 290: it should be “Gorospe [42] analysed”

Author Response

Dear teacher,

Thank you for your feedback and allow us to improve our manuscript "Human health risk distribution and safety threshold of cadmium in soil of coal chemical industry area",(minerals-1268382). Next, we respond to the reviews of reviewer. The reviewer' comments are in black text, and our brief reply is in red text; the newly revised part of our reply is in blue text, and the unchanged original text is black. This is consistent with the newly revised manuscript.

The modification details are as follows:

  1. Lines 93-94: the following sentence is unclear: “By the Kriging interpolation method and process production line, the evaluation results were further processed..” – please rephrase

Reply: Thank you very much for your comments. After reading the manuscript carefully, to ensure the rigor of the manuscript, we had revised the details.

New revision: see "Introduction" on P3, L95 of the revised manuscript.

The Kriging interpolation method was used to deeply process the evaluation results. On this foundation, combined with the process production line, and then the distribution map of human health risk assessment in the plant area was obtained, so that the Cd hazards in different production units were accurately evaluated and their risk sources were analyzed.

  1. Line 100: please insert “on” between “Cd” and “human health”

Reply: Thank you very much for your reminding. We had revised the details.

New revision: see "Introduction" on P3, L103 of the revised manuscript.

The reasonable and accurate evaluation of the pollution degree and health risk can provide an effective and beneficial reference value for land planning and utilization, and provide a theoretical basis for reducing the harm of Cd on human health in coal chemical sites for further study.

  1. Line 107: please be more precise in the following statement “The surface water around the plant area is poor” – what does “poor water” mean?

Reply: Thank you very much for your comments. The meaning of this sentence is that there is a lack of surface water resources. After reading the manuscript carefully, in order to express more accurately, we modified this sentence.

New revision: see "Study area" on P3, L110 of the revised manuscript.

The surface water around the plant area is deficient.

  1. Figure 1. the quality of this figure have to be improved, mainly in the part showing the regional distribution map of the plant - it is currently difficult to read.

Reply: Thank you very much for your comments on Figure 1. After careful consideration, we modified Figure 1.

New revision: see "Study area" on P4, L126 of the revised manuscript.

  1. Lines 129-130: this sentence seems to be unfinished “Offset collection or skipping of locations that are difficult to collect, such as pools, workshops, and slag dumps” – please rephrase

Reply: In the newly revised manuscript, this part had been revised.

New revision: see "Soil sampling and analysis" on P4,L133 of the revised manuscript.

In the process of sampling, some sampling points located in areas such as pools, workshops, slag piles, and other areas, which are not conducive to soil collection. Make appropriate adjustments according to the actual environment around the preset sampling points.

  1. Lines 151-189: Subchapter “Human health risk assessment”: in this chapter both the models and parameters used for carcinogenic and non-carcinogenic risk calculation should be given (in methodological part or in supplementary materials)

Reply: Thank you very much for your reminding. We had added the response formula to the new revision.

New revision: see "Human health risk assessment" on P6-8 of the revised manuscript.

  1. Lines 179-181: please carefully check (and correct) explanations for Ri and CRi - the explanations seem to be identical

Reply: Thank you very much for your reminding. In the newly revised manuscript, we had revised the details.

New revision: see "Risk characterization and contribution rate" on P8, L198 of the revised manuscript.

In the above formula: CRi means the carcinogenic risk contribution rate or hazard quotient level of a certain exposure route, and the dimension is 1; ΣCRi means the total carcinogenic risk or total hazard quotient.

  1. Lines 191-199: please indicate which statistical tool was used for the kriging analysis and statistical evaluation of Cd content in soils

Reply: In the newly revised manuscript, this part had been revised.

New revision: see "Kriging interpolation method" on P9, L215 of the revised manuscript.

The Kriging interpolation method of the GIS spatial model is a geostatistical method used to smooth surfaces and to predict the values of unsampled locations.

  1. Line 219: it should be “It can be seen from Table 2 that the value of Morans’ I was…”

Reply: Thank you very much for your reminding. In the newly revised manuscript, we had revised the details.

New revision: see "Spatial autocorrelation test" on P9, L245 of the revised manuscript.

It can be seen from Table 6 that the value of Morans’ I was relatively close to 1, so the hazard quotient of Cd was a significant cluster with a positive correlation [35].

  1. Lines 221-222: please change “Table 1” to “Table 2”

Reply: Thank you very much for your reminding. In the newly revised manuscript, we had revised the details.

New revision: see "Spatial autocorrelation test" on P9, L248 of the revised manuscript.

By the criterion of spatial autocorrelation, compared with z value and p value in Table 6.

  1. Line 232: the results of semi-variance model are given in Table 3 (not in Table 6).

Reply: Thank you very much for your reminding. In the newly revised manuscript, we had revised the details.

New revision: see "Analysis of spatial structure variation" on P10, L258 of the revised manuscript.

Table 7 showed the fitting results of the semi-variance model of total carcinogenic risk and total hazard quotient of Cd in Kriging interpolation points.

  1. Line 234: It should be “the ratio of nugget value C0to the sill value (C0/Sill)”.

Reply: Thank you very much for your reminding. In the newly revised manuscript, we had revised the details.

New revision: see "Analysis of spatial structure variation" on P10, L261 of the revised manuscript.

From the ratio of nugget value C0 to the sill value (C0/Sill), the interpolated value of Cd was ranged from 33.3% to 34.64%.

  1. Line 244: please change “consequences” to “results”

Reply: Thank you very much for your reminding. In the newly revised manuscript, we had revised the details.

New revision: see "Human health risk assessment of soil Cd in coal chemical area" on P10, L271 of the revised manuscript.

The results are shown in Table 8.

  1. Table 4. Please specify what is given in this Table – minimum/maximum or mean/median values

Reply: Thank you very much for your reminding. In the newly revised manuscript, we had revised the details.

New revision: see "Human health risk assessment of soil Cd in coal chemical area" on P10, L273 of the revised manuscript.

  1. Lines 282-283: “(with an average of 0.020)” – mean values are not given in Table 4 and Figure 4 – please correct results

Reply: Thank you very much for your reminding. In the newly revised manuscript, we had revised the details.

New revision: see "Human health risk assessment of soil Cd in coal chemical area" on P12, L313 of the revised manuscript.

  1. Figure 3 and 4: text in both Figures is blurry and unreadable – please improve

Reply: Thank you very much for your reminding. In the newly revised manuscript, we had revised the details.

New revision: see "Carcinogenic risk" and "Hazard quotient" P11-12 of the revised manuscript.

  1. Line 290: it should be “Gorospe [42] analysed”

Reply: Thank you very much for your reminding. In the newly revised manuscript, we had revised the details.

New revision: see "Hazard quotient" P13, L321 of the revised manuscript.

Gorospe [47] analyzed 16 different heavy metals in 91 soil samples from a vegetable garden in San Francisco.

Yours sincerely,

Kai Zhang

Name: Kai Zhang

E-mail: zhangkai@cumtb.edu.cn

Round 2

Reviewer 2 Report

Previously, I had several comments, the authors addressed all of them and made appropriate changes to the manuscript. The methodological part is now sufficiently detailed, the models and parameters used for risk calculation were included in the “Materials and Methods”. The unclear statements and mistakes in were corrected and the quality of Figures was also improved as suggested.
However, I suggest minor revision – text editing is needed mainly in parts which were changed after revision. In my opinion, the corrected parts of the manuscript (marked in red in the new version) require linguistic correction because they are difficult to understand, e.g. lines 130-132, text in Table 7 - coefficient of determination.

Author Response

Dear teacher,

Thank you for your feedback and allow us to improve our manuscript "Human health risk distribution and safety threshold of cadmium in soil of coal chemical industry area",(minerals-1268382). Based on your revised comments, we had carefully read the revised part of the manuscript marked in red and made further revisions.

The modification details are as follows:

  1. Lines 132-135:

In the process of sampling, some sample points in pools, workshops, slag piles, and other areas, which are detrimental to soil collection. Therefore, it is necessary to make appropriate adjustments according to the actual environment around the preset sampling points.

  1. Lines 172-174:

The soil exposure model (Table 1) and exposure factor parameters (Table 2) corresponding to carcinogenic and non-carcinogenic effects of a single pollutant were selected [25].

  1. Lines 214-215:

The Kriging interpolation of the GIS spatial model is a geostatistical method, which is used in smoothing surfaces and predicting the values of unsampled locations. 

  1. Table 7:

Coefficient of determination/R2

Thank you again for your attention to our manuscript.

Yours sincerely,

Kai Zhang

Name: Kai Zhang

E-mail: zhangkai@cumtb.edu.cn

This manuscript is a resubmission of an earlier submission. The following is a list of the peer review reports and author responses from that submission.

Round 1

Reviewer 1 Report

Cd is certainly one of the most toxic elements, but it is not the only one. So I wonder why the authors focused on Cd only, without evaluating the overall metal pollution of the research area.

All the exponential values should be reported with a correct scientific notation, instead that with the E-notation.

Specific comments:

Line 47: A situation cannot be optimistic: it can be either good or bad, worrying or not…

Lines 52-53: It deeply depends on the considered land! It is not true in any situation.

Line 78: As I noted before, Cd is not the main pollutant of EVERY heavy metal polluted soil. So, please change.

Line 107: Please explain what zone A is.

Lines 107-110: I suppose these values were not registered by the authors, so please cite.

Line 125: Please re-formulate this sentence.

Lines 127-129: This is not an explanation of how the process was performed, yet of how it should be performed. Moreover, information such as “sample name and code should never be changed” is useless, since it is something which applies to each well-done laboratory work.

Lines 130-132: Soil samples were probably acid-digested prior to analysis. Please explain in detail, reporting which acids were used, in which quantities and which method was used for accelerating the reaction (e.g. microwave).

Line 138: The study was performed in the past, so please do not use future tenses.

Lines 146-150: Please check grammar.

Line 193: Please correct: the results ARE shown.

Line 196 and table 1: Please be coherent: P-value and Z-value or P value and Z value. Moreover, P and Z should not be capitalized.

Lines 196-198: Please check grammar.

Table 1: As far as I know, a p-value cannot be equal to zero. If the p-value is too small to be reported, I suggest indicating <0.001

Lines 203-206: Please check grammar.

Lines 211-213: Please cite relevant literature.

Line 218: The correct expression is “The descriptive statistics of Cd”

Line 219: ARE listed

Line 225: Nothing is obvious! It might be simply due to the natural variability. So please change the sentence in order to highlight it is simply a hypothesis.

Line 243: Is this difference statistically significant?

Lines 247-248: Please do not use future tenses, since it has no sense.

Line 264: Cd is NOT a volatile element. However, extremely small soil/dust particles rich of Cd might actually be volatile and reach the respiratory system. Please correct.

Figure 3: I really do not understand on which basis the authors state that inhalation mainly determine a non-carcinogenic risk (and what does this mean in detail?) while oral intake and skin contact mainly determine a carcinogenic risk.

Author Response

Dear teacher,

Thank you for your feedback and allow us to improve our manuscript "Human health risk distribution and safety threshold of cadmium in soil of coal chemical industry area", (minerals-1200224). Next, we respond to the reviews of the reviewer. The reviewer's comments are in black text, and our brief reply is in red text; the newly revised part of our reply is in blue text, and the unchanged original text is black. This is consistent with the newly revised manuscript.

Thanks again for the reviewer's comments that helped us improve our paper.

Specific comments:

  1. Line 47: A situation cannot be optimistic: it can be either good or bad, worrying or not…

Reply: Thank you very much for your comments. We didn't describe the manuscript accurately enough. In the revised manuscript, we had revised it as "the overall pollution situation is relatively serious".

New revision: see "Introduction" on P2, L48 of the revised manuscript.

  1. Lines 52-53: It deeply depends on the considered land! It is not true in any situation.

Reply: In response to your comments, we had carefully read the description of the sentence in the manuscript. In order to ensure the preciseness of the manuscript, we had deleted it from the original text.

New revision: see "Introduction" on page 2 of the revised manuscript.

  1. Line 78: As I noted before, Cd is not the main pollutant of EVERY heavy metal polluted soil. So, please change.

Reply: Thank you very much for your comments, we had modified it in the newly revised manuscript.

New revision: see "Introduction" on P2, L77-78 of the revised manuscript.

It is a great challenge to accurately simulate the spatial distribution of Cd in soil due to its complex pollution causes.

  1. Line 107: Please explain what zone A is.

Reply: Zone A here refers to the production area in the coal chemical plant area.

  1. Lines 107-110: I suppose these values were not registered by the authors, so please cite.

Reply: Thank you very much for your reminder. We had added the reference to the references.

New revision: see "References" on P12, L424 of the revised manuscript.

http://data.cma.cn/data/cdcdetail/dataCode/SURF_CLI_CHN_MUL_DAY_640.html

  1. Line 125: Please re-formulate this sentence.

Reply: In the newly revised manuscript, the sentence had been re-formulated, making the article more rigorous and scientific.

New revision: see "Soil sampling and analysis" on P3, L123 of the revised manuscript.

To comprehensively and accurately evaluate the pollution degree of soil heavy metals at different locations in the plant, the 50m×50m checkerboard distribution method was used to collect soil samples in this study.

  1. Lines 127-129: This is not an explanation of how the process was performed, yet of how it should be performed. Moreover, information such as “sample name and code should never be changed” is useless, since it is something which applies to each well-done laboratory work.

Reply: Thank you for your comments. This sentence exists because we considered the rigor of the test when we compiled the manuscript, so we made this description. We had deleted it in the original text.

New revision: see "Soil sampling and analysis" on P4 of the revised manuscript.

  1. Lines 130-132: Soil samples were probably acid-digested prior to analysis. Please explain in detail, reporting which acids were used, in which quantities and which method was used for accelerating the reaction (e.g. microwave).

Reply: Thank you very much for your comments on our manuscript, we had added the details of the test.

New revision: see "Soil sampling and analysis" on P4 of the revised manuscript.

0.300g sample was accurately weighed and put into a beaker. 10 ml concentrated HNO3, 4ml 3% (mass fraction) H2O2, and 10 ml HF were added in turn for microwave digestion. After digestion, 1% (mass fraction) HNO3 was added to a 50 ml colorimetric tube for standby.

  1. Line 138: The study was performed in the past, so please do not use future tenses.

Reply: In the newly revised manuscript, the tense had been unified, which made the language of the article richer and more accurate.

New revision: see "Quality Control" on P4, L142-143 of the revised manuscript.

  1. Lines 146-150: Please check grammar.

Reply: In the newly revised manuscript, we had revised the grammar of the sentence.

New revision: see "Human health risk assessment" on P5 of the revised manuscript.

The Ministry of Environmental Protection of the PRC released the Technical guidelines for risk assessment of soil contamination of land for construction [21]. The human health risk assessment models recommended in the technical guidelines are based on the EPA models. However, in order to reflect the actual contamination situations in China, unique parameters are given according to the environment and living habits of Chinese residents.

  1. Line 193: Please correct: the results ARE shown.

Reply: In the newly revised manuscript, we revised the details to make the structure of the article clearer.

New revision: see "Model usage conditions " on P6, L199 of the revised manuscript.

  1. Line 196 and table 1: Please be coherent: P-value and Z-value or P value and Z value. Moreover, P and Z should not be capitalized.

Reply: In the newly revised manuscript, we had revised the details.

New revision: see "Model usage conditions" on page 6 of the revised manuscript.

  1. Lines 196-198: Please check grammar.

Reply: In the newly revised manuscript, we had revised the grammar of the sentence.

New revision: see "Model usage conditions" on page 6 of the revised manuscript.

The spatial autocorrelation (Moran I) tool of ArcGIS spatial statistics was used to analyze the carcinogenic risk and hazard quotient of Cd pollution.

  1. Table 1: As far as I know, a p-value cannot be equal to zero. If the p-value is too small to be reported, I suggest indicating <0.001

Reply: Thank you very much for your comments, we had modified it.

New revision: see "Model usage conditions" on Table 1 page 6 of the revised manuscript.

  1. Lines 203-206: Please check grammar.

Reply: In the newly revised manuscript, we had revised the grammar of the sentence.

New revision: see "Model usage conditions" on page 6 of the revised manuscript.

  1. Lines 211-213: Please cite relevant literature.

Reply: Thank you very much for your reminder. We had added the literature to the references.

New revision: see "References" on P13, L437 of the revised manuscript.

Guan,Y.J.; Zhou,W.; Bai, Z.k.; Cao,Y.G.; Huang,Y.H.; Huang,H.Y. Soil nutrient variations among different land use types after reclamation in the Pingshuo opencast coal mine on the Loess Plateau, China. Catena. 2020, 188.

  1. Line 218: The correct expression is “The descriptive statistics of Cd”

Reply: In the newly revised manuscript, this part had been revised.

New revision: see "Results and discussion" on P6 of the revised manuscript.

  1. Line 219: ARE listed

Reply: In the newly revised manuscript, we had revised the details.

New revision: see "Results and discussion" on P6 of the revised manuscript.

  1. Line 225: Nothing is obvious! It might be simply due to the natural variability. So please change the sentence in order to highlight it is simply a hypothesis.

Reply: Thank you very much for your comments. After reading the manuscript carefully, to ensure the rigor of the manuscript, we decided to delete this sentence.

New revision: see "The descriptive statistics of Cd" on P7 of the revised manuscript.

  1. Line 243: Is this difference statistically significant?

Reply: Thank you very much for your comments. Here we divided the carcinogenic risk into two parts. The main starting point was that the color difference between high-value areas and low-value areas was relatively obvious, which made the significance of the article more prominent.

  1. Lines 247-248: Please do not use future tenses, since it has no sense.

Reply: In the newly revised manuscript, this part had been revised.

New revision: see "Carcinogenic risk" on P7 of the revised manuscript.

  1. Line 264: Cd is NOT a volatile element. However, extremely small soil/dust particles rich of Cd might actually be volatile and reach the respiratory system. Please correct.

Reply: In the newly revised manuscript, this part had been revised.

New revision: see "Carcinogenic risk" on P8 of the revised manuscript.

  1. Figure 3: I really do not understand on which basis the authors state that inhalation mainly determine a non-carcinogenic risk (and what does this mean in detail?) while oral intake and skin contact mainly determine a carcinogenic risk.

Reply: Thank you very much for your comments. In our study, we found that the concentration of Cd in this area exceeded the standard, so this manuscript explored the human health risk assessment of Cd in this area. According to the calculation results, we found that in our study area, the main way of non-carcinogenic risk was inhalation, while the relative value of the other two ways was low. By calculation, it was found that the main pathways of carcinogenic risk were oral and skin contact (see Figure 5).

Yours sincerely,

Kai Zhang

Name: Kai Zhang

E-mail: zhangkai@cumtb.edu.cni

Reviewer 2 Report

Review of the article "Human health risk distribution and safety threshold of cadmium in soil of coal chemical industry area"

Authors (Kai Zhang, Xiaonan Li, Zhenyu Song, Jiayu Yan, MengYue Chen and Juncheng Yin) have demonstrated their knowledge of the subject. They skilfully selected the problem and the research area. They described interesting studies that are worth publishing.

Please explain:

  1. Is the coal chemical plant a source of only metals Cd or also other heavy?
  2. Table 4, how is overall different from N = 153, from how many samples and how is overall calculated?
  3. Please provide the date (year, month) of sampling and collection sites, adjusting them to Figures 1 and 2.
  4. Please present figures 1 and 2 in the same size and increase the readability of marked fields. Which field had the most samples? The methodology should refer to the designation used in these figures (e.g. wasterwater treatment unit) and provide the number of samples in each area. Please also use the appropriate scale for figures 1 and 2.
  5. Please complete the caption for figure 2.
  6. Line 258, please use a space before quoting the literature, and similarly for the whole text, please add spaces in appropriate places.
  7. Line 299, please write 4 and 73 times, similarly line 302 rounding a percentage to one decimal point.
  8. Please standardize the name of the element throughout the text, or write it all or the symbol Cd.
  9. Figure 3 how was non-carcinogenic risk calculated?
  10. Lines 345-347 should be given more than once instead of a value.

Greeting,

Reviewer

Author Response

Dear teacher,

Thank you for your feedback and allow us to improve our manuscript "Human health risk distribution and safety threshold of cadmium in soil of coal chemical industry area", (minerals-1200224). Next, we respond to the reviews of the reviewer. The reviewer's comments are in black text, and our brief reply is in red text; the newly revised part of our reply is in blue text, and the unchanged original text is black. This is consistent with the newly revised manuscript.

Thanks again for the reviewer's comments that helped us improve our paper.

This manuscript deal with the Human health risk assessment and spatial distribution of fluoride from shallow groundwater in a region of southwest China. The results are meaningful. I think the manuscript could be accepted with minor revise.

1 Is the coal chemical plant a source of only metals Cd or also other heavy?

Reply: Thank you for your comments on the manuscript. Coal chemical plant area is not only the source of Cd but also other heavy metals, such as arsenic, chromium, lead, and mercury. In our manuscript, we tested and analyzed the types and concentrations of heavy metals in the soil samples after sampling, and found that the pollution of Cd in the study area was relatively high, and the pollution characteristics were obvious. In addition, Cd has health hazards to the human body. In conclusion, in order to explore the health risks of Cd in this area, we studied it.

2 Table 4, how is overall different from N = 153, from how many samples and how is overall calculated?

Reply: There were 153 sampling points in our manuscript, so 153 sampling points were referred to as a whole, the maximum and minimum values were taken as the range of pollutants, and the maximum and minimum values were taken as the risk assessment results of Cd pollution in different coal gasification areas.

3 Please provide the date (year, month) of sampling and collection sites, adjusting them to Figures 1 and 2.

Reply: Thank you for your comments on the manuscript. The date of sampling and collection sites had been revised in our manuscript. In addition, Figure 1 and Figure 2 were added to the revised manuscript to make the structure of the article more clear and complete.

New revision: see "Study area" on page 3 and 4 of the revised manuscript.

4 Please present figures 1 and 2 in the same size and increase the readability of marked fields. Which field had the most samples? The methodology should refer to the designation used in these figures (e.g. wasterwater treatment unit) and provide the number of samples in each area. Please also use the appropriate scale for figures 1 and 2.

Reply: Thank you for your comments on the manuscript. Figure 1 and Figure 2 were added to the revised manuscript.

New revision: see "Study area" on page 3 and 4 of the revised manuscript.

Figure 1. Location of the study area

Figure 2. Distribution of sampling points

5 Please complete the caption for figure 2.

Reply: Thank you very much for your comments on our manuscript. We had added to the revised manuscript

Figure 4. Total hazard quotient risk map for Cd

New revision: see "Results and discussion" on page 9 of the revised manuscript.

6 Line 258, please use a space before quoting the literature, and similarly for the whole text, please add spaces in appropriate places.

Reply: Thank you for your comments. Spaces had been added before the citation of the full text to make the structure and language of the article more rigorous.

New revision: see the full text of the revised manuscript.

7 Line 299, please write 4 and 73 times, similarly line 302 rounding a percentage to one decimal point.

Reply: In the newly revised manuscript, the expression had revised.

New revision: see "Results and discussion" on P9, L302 of the revised manuscript.

However, for the non-carcinogenic risk of Cd in the coal chemical industry area, the contribution rate of the three exposure pathways of Cd in the soil of the coal chemical industry area was respiratory inhalation (79%)> oral intake (17%)> skin contact (4%).

8 Please standardize the name of the element throughout the text, or write it all or the symbol Cd.

Reply: In the newly revised manuscript, the names of elements in the whole text had been standardized and written as Cd.

New revision: see the full text of the revised manuscript.

9 Figure 3 how was non-carcinogenic risk calculated?

Reply: The non-carcinogenic risk value was calculated by the following formula.

Table 1. Calculating models of soil exposure doses in three exposure routes

Exposure

routes

Explanation

Formula expression of exposure

Oral intake

Carcinogenic

Non-carcinogenic

Skin contact

Carcinogenic

Non-carcinogenic

Inhalation in soil particle

Carcinogenic

Non-carcinogenic

10 Lines 345-347 should be given more than once instead of a value.

Reply: Thank you very much for your comments. We had selected the maximum value as the risk threshold under different approaches to more intuitively represent the safety threshold of cadmium in the regional soil.

Yours sincerely,

Kai Zhang

Name: Kai Zhang

E-mail: zhangkai@cumtb.edu.cni

Reviewer 3 Report

The manuscript deals with the topic of soil contamination by Cd. This topic is frequent generally nevertheless the manuscript focus on the coal mining area in China where the environmental problems connected with coal mining and combustion are known. The evaluation of Cd human health risks based on carcinogenic and non-carcinogenic risks is suitable tool for presented study. It must be appreciated the the authors did not used generally recommended values of Cd in soil only and applied national and regional background values in soil. This approach showed some specific problems in the area and some practical recommendations are following from the study. The evaluation of soil contamination by Cd in area of interest was done correctly. The manuscript can be be use and an example pattern for other areas loaded by coal processing. The formal level of manuscript is high and I did not find any technical problems.    

Author Response

Dear teacher,

Thank you very much for your affirmation of our manuscript"Human health risk distribution and safety threshold of cadmium in soil of coal chemical industry area", (minerals-1200224)., and we will continue to work hard. We will do more in-depth research in this field in the future, in order to provide theoretical guidance for the determination of heavy metal safety threshold in China's coal chemical industry.

Yours sincerely,

Kai Zhang

Name: Kai Zhang

E-mail: zhangkai@cumtb.edu.cni

Reviewer 4 Report

In my opinion, this article should not be accepted for publication in this journal. As it is claimed in the manuscript, previous works related to the same project were published in other journals. For example, reference 22 (Zhang,K.; Qiang,C.D.; Liu,J. Spatial distribution characteristics of heavy metals in the soil of coal chemical industrial area. J. Soils Sediments, 2018, 18(5),2044-2052) reveals the sampling campaign and geochemical results. Specifically, Cd concentrations distribution are spatially represented. In this manuscript, health risk for Cd is also represented, revealing the same pattern. This is the expected result since the health risk calculation is directly proportional to Cd concentration. In my opinion, this work did not reveal any new scientific contribution. It is true that, although Cd concentrations are not higher than the Soil Screening Levels, health risk values are higher than the acceptable levels, but the work sounds like a technical report in spite of a scientific article. Consequently, I think that the manuscript should be rejected.

Author Response

Dear teacher,

Thank you for your feedback and allow us to improve our manuscript "Human health risk distribution and safety threshold of cadmium in soil of coal chemical industry area", (minerals-1200224).

The reason why we separately assessed the human health risk of soil cadmium in the area was that we detected the heavy metal type and concentration in the soil sample and found that the soil cadmium exceeded the standard, in order to further explore its human health risk. And to further describe its pollution, we wrote the manuscript. With a view to effective management and control of soil cadmium in this area.

Yours sincerely,

Kai Zhang

Name: Kai Zhang

E-mail: zhangkai@cumtb.edu.cni

Reviewer 5 Report

Revision of the article: Human health risk distribution and safety threshold of cadmium in soil of coal chemical industry area.

Dear authors, thanks for the effort to write this paper, however I consider that it should be reject. A new submission after a deep modification in the paper should be done.

Major concerns:

The language is below the par, it should be polished carefully.

Introduction has not well connected and methodology is poorly described. Results and discussion, as well as, the figure presentation and description need be re-thinking.Also there is several information included in the manuscript that was previously published by the authors in other research paper (Spatial distribution characteristics of heavy metals in the soil of coal chemical industrial area). For example, all the data of the table 1, table 2 and table 3 are in this paper published too. I think that the paper could be more interesting if all the elements studied in the previous paper (Pb, Hg, Cd, Cr and As) , where the spatial distribution was study, are considered now for the health risk assessment, and not only Cd.

Some minor comments:

L15: Change order: "to evaluate the human health risk of Cd by Kriging interpolation method".

L17.Give range of average, not both and I suppose that you want says that was 4.8 to 5.56 times HIGHER

L35. China. Add the dot.

L59. Of which mining area?

Figure 1. The information of the location is OK but the square at the right side is completely unclear and not described in the text and in the caption of the figure. I think that if there is a picture of the plant it could be easily to describe, because the plant distribution appears also later in figure 2.

Add: Figure 2. Distribution of the sampling points "in the coal plant".

From my point of view, It is a bit weird that there are sampling points over regions that seem to be constructions.

L212. "of OK"?

L220-221. It is a result not methodology.

L223 and so on. They are results not methodology

What are nugget and sill values?

L252. The Kriging

Conclusion and abstract section repeat a lot of information even with the same words.

Round 2

Reviewer 4 Report

I reiterate my last decision. I think the manuscript should be rejected.